# A Novel Oncogenic Role of FDX1 in Human Melanoma Related to PD-L1 Immune Checkpoint

**DOI:** 10.3390/ijms24119182

**Published:** 2023-05-24

**Authors:** Huijiao Lu, Jiahua Liang, Xue He, Huabin Ye, Chuangdong Ruan, Hongwei Shao, Rongxin Zhang, Yan Li

**Affiliations:** 1Department of Biotechnology, School of Life Sciences and Biopharmaceutics, Guangdong Pharmaceutical University, Guangzhou 510006, China; 2Department of Pancreato-Biliary Surgery, The First Affiliated Hospital, Sun Yat-sen University, Guangzhou 510080, China

**Keywords:** FDX1, tumor immune microenvironment (TME), immune checkpoint genes (ICPs), anti-tumor immunotherapy, flow cytometry (FCM), PD-L1

## Abstract

The aim of this study was to evaluate the association between Ferredoxin 1 (FDX1) expression and the prognostic survival of tumor patients and predict the efficacy of immunotherapy response to antitumor drug sensitivity. FDX1 plays an oncogenic role in thirty-three types of tumors, based on TCGA and GEO databases, and further experimental validation in vitro was provided through multiple cell lines. FDX1 was expressed highly in multiple types of cancer and differently linked to the survival prognosis of tumorous patients. A high phosphorylation level was correlated with the FDX1 site of S177 in lung cancer. FDX1 exhibited a significant association with infiltrated cancer-associated fibroblasts and CD8^+^ T cells. Moreover, FDX1 demonstrated correlations with immune and molecular subtypes, as well as functional enrichments in GO/KEGG pathways. Additionally, FDX1 displayed relationships with the tumor mutational burden (TMB), microsatellite instability (MSI), DNA methylation, and RNA and DNA synthesis (RNAss/DNAss) within the tumor microenvironment. Notably, FDX1 exhibited a strong connection with immune checkpoint genes in the co-expression network. The validity of these findings was further confirmed through Western blotting, RT-qPCR, and flow cytometry experiments conducted on WM115 and A375 tumor cells. Elevated FDX1 expression has been linked to the enhanced effectiveness of PD-L1 blockade immunotherapy in melanoma, as observed in the GSE22155 and GSE172320 cohorts. Autodocking simulations have suggested that FDX1 may influence drug resistance by affecting the binding sites of antitumor drugs. Collectively, these findings propose that FDX1 could serve as a novel and valuable biomarker and represent an immunotherapeutic target for augmenting immune responses in various human cancers when used in combination with immune checkpoint inhibitors.

## 1. Introduction

FDX1, a universal protein, participates in many biological processes as an electron donor and has a critical role in iron–sulfur (Fe-S) cluster biosynthesis with electron transfer activity. The mammalian mitochondria, FDX1 (also known as adrenodoxin, Adx), is involved in bile acid (BA) synthesis and adrenal steroidogenesis [1,2,3]. Its electrons are delivered from NADPH to FDX1 by ADX reductase (FDXR or ADR) via biochemical pathways [4,5]. Members of the ferredoxin family, including human mitochondrial ferredoxin 1 (FDX1) and FDX2 (also known as FDX1L), are identified as having distinct and substrate-specialty activities according to protein sequence alignment, which indicated the predicted hypothesis that mature FDX1 has 43% and 69% similarity with FDX1L. FDX1 and FDX2 also displayed dramatically various specificity for diverse biochemical processes, although they had remarkable sequence similarity. Additionally, FDX1 is involved in some metabolic processes such as steroid hormone synthesis as the mitochondria reductase in mammalian tissues. As a member of the 2Fe-2S cluster ferredoxin family, FDX2 transferred electrons from NADPH to FDX1 by FDXR in the mitochondria and played a key role in heme A and Fe-S protein biosynthesis [6,7,8,9,10]. It has been reported that, in the ferredoxin family, with its redox specialty, the ferredoxin residues close to the iron–sulfur cluster of FDX1, including its oxidized and reduced stages, participated in the interaction with protein complexes such as those containing cysteine desulfurase (NFS1) and ISD11 (also named LURM4). Furthermore, members of the subfamily [2Fe-2S] and [4Fe-4S] consist of [Fe-S] proteins and transfer electrons in distinct biochemical reactions through various species whose [Fe-S]-binding motifs are typically part of the mitochondrial metabolism [11,12]. Their family derives from bacteria with a molecular mass of 6-25kDa and is well known in mitochondria and plastids of algae and plants [11,13,14,15,16]. Emerging publications displayed that FDX1, an essential biomarker for cuproptosis, is involved in the tricarboxylic acid (TCA) cycle and mitochondrial metabolism to enhance adaptation to proteotoxic stress, which leads to cell death in the mitochondria. FDX1 expression and mitochondria-dependent energy metabolism induce copper-dependent cell death [17,18,19]. As an essential component, copper interferes with iron and sulfur clusters (Fe-S), leading to greater sensitivity to copper-mediated cell death [20,21,22,23,24]. Copper-binding ionophores such as elesclomol with a high FDX1 level could bind with copper and then deliver it to the mitochondria, producing a higher concentration of reactive oxygen species (ROS), increased by inducing Cu (I) into Cu (II) [25,26,27,28,29,30,31]. Furthermore, oxidative phosphorylation promotes increased sensitivity to FDX1-regulated copper-dependent cell death. Moreover, it has been reported that abnormal copper accumulation or copper-induced DNA damage might cause neurodegenerative disease.

It has also been reported that constant inflammation is a hallmark of tumors, which might promote tumor growth or tumorigenesis [32]. The entire immune landscape changes due to tumorous development and progression. The tumor microenvironment (TME) consists of a tissue extracellular matrix (ECM) and stromal fibroblast cells, which are combined with stromal infiltration cells, forming the tumor purity that relies on the estimation of stromal cells and immune cells in cancerous samples, as examined in the TCGA project [33,34,35]. Moreover, distinct stromal factors such as fibroblast cells operate as mediators of cancerous stromal cells’ dense communication to integrate with the TME. Stromal cells retain their activated state to promote tumor progression and aggressiveness, due to the crosstalk both in tumor and stromal cells. Immune cells within the TME foster tumor progression in the tumorous immunosuppressive microenvironment (TIM) [36,37], which functions as an immunosuppressive component [38]. Meanwhile, the diverse tumorous cellular subsets, such as stromal, tumor, and infiltrating immune cells, interact and promote the construction of a highly immunosuppressive microenvironment. In particular, these cancer-related fibroblast (CAF) cells and CAF-based components operate as direct tumorous survival signals, which mainly inhibit the activity of immune effector cells and recruit much more immunosuppressive cells, leading to the tumor cells escaping immune surveillance [39,40,41]. Genetic mutation and tumor mutational burden (TMB)/microsatellite instability (MSI) can produce a neoantigen that is recognized by effector T cells, which is also seen as an emerging predictive biomarker for immunotherapy. TMB and non-synonymous somatic mutations, including single-nucleotide variants and small insertions or deletions, represent the tumor genome stability and heterogeneity within the TME. MSI was characterized as mutations in the insertion or deletion/indel within the repetitive sequence stretches [42,43,44]. Some evidence reported that high TMB is closely related to the immune checkpoint blockade (ICB) in the immunotherapy response of cancerous patients [45,46,47]. It was also reported that TMB is closely positively related with the prognostic survival of the block of immune checkpoint inhibitors (ICIs), which could reactivate immune cell functions to kill and eliminate tumor cells. In recent years, anticancer immunotherapy focusing on the immune system has become increasingly critical in tumorous therapy. Moreover, immunotherapy aims to stimulate the immune system and activate immune cells to recognize tumorous surface antigens as well clear up cancer cells in order to contribute to antitumor immune treatment.

Immune checkpoint molecules (ICPs) such as PD-L1 and CTLA-4 modulate the immune system of patients to maintain remission in multiple types of cancer. PD-L1, an immune checkpoint protein, mainly presents in tumorous infiltration immune cells and cancerous cells in the TME and serves to inhibit various immune cells and invaders from immune attacks to attain anticancer immune responses. However, cancerous cells and immune cells within the TME generate IFNs to suppress the activity of immune cells and foster the efficacy of antitumor immunotherapy; as we know, IFN-II (also known as IFN-γ) plays a critical role in innate and adaptive immunity and performs regulatory immune activities. Certain immune cells, including CD8^+^ T cells, macrophages, monocytes, and eosinophils, are closely correlated with immune functions in the MHC family, and eventually present antigens to cytotoxic CD8^+^ T cells. It has been reported that drugs that reduce cholesterol could suppress PD-L1 expression, which plays a critical role in fostering tumor cells’ immune invasion [48,49]. A high level of cholesterol promotes tumor cells to proliferate and invade rapidly. An increasing level of cholesterol within the TME could impact the activity of tumorous infiltration-immune cells. Recently, increasing evidence reported that cholesterol binding to the transmembrane domain of PD-L1 might make PD-L1 expression stable in tumor cells, which offers an effective way to avoid PD-L1-mediated tumorous immune escape. The regulation of cholesterol has been examined as a logical strategy to enhance anticancer immunotherapy efficacy. Cholesterol is an important ingredient for the body’s synthetic steroid hormones. It also has an essential role in tumorous development and progression as a key functional molecule for cell survival and structure. Cholesterol levels in unaffected cells are closely regulated at multiple stages to maintain healthy physiological functions. However, cholesterol levels in tumor cells often remain at a high level, and cholesterol metabolism is extremely active in meeting the needs of rapid tumor cell proliferation, invasion, and metastasis.

As reported, elesclomol has undergone clinical trials for the diagnosis of various conditions. Tumors characterized by high FDX1 levels have shown potential responsiveness to elesclomol. However, these trials still lack key biomarkers for identifying tumor patients and a clear understanding of the concrete mechanisms underlying the drug’s effects. Therefore, a comprehensive and systematic analysis of FDX1 across multiple cancer types was conducted. Given the heterogeneity and complexity of tumorigenesis, it is critical to explore the expression patterns of FDX1 in a multicancer context and assess its association with clinical prognostic survival as well as potential molecular mechanisms. Analytical data were primarily collected from publicly funded TCGA, XENA, and NCBI databases. The analysis initially focused on the TCGA and GEO datasets, enabling a multicancer examination of FDX1. Subsequently, several components were integrated, including gene expression profiles, survival prognosis stages, genetic mutations, immune infiltration-related functions, GO/KEGG pathway enrichment, and drug sensitivity. These components were incorporated to investigate the potential molecular mechanisms through which FDX1 contributes to the pathogenesis and physiological prognostic survival of diverse tumors.

## 2. Results

### 2.1. The Flow Chart of Multicancer Expression Analysis of FDX1

The potential oncogenic role of FDX1 in 33 types of tumors was explored using TCGA, GEO, and XENA datasets. The protein structure of FDX1 is conserved across different species, such as H. sapiens, P. troglodytes, M. mulatta, C. lupus, and B. taurus (Appendix A). Additionally, phylogenetic tree data demonstrate the evolutionary correlation of FDX1 among diverse species (Appendix A). A high expression of FDX1 is significantly associated with the survival prognosis of patients across multiple cancers. Genetic mutations play a vital role in antitumor immunity.

Through GO/KEGG enrichment analysis, we identified that FDX1 may be involved in the regulation of cancer through pathways related to “cancers” and “metabolic pathways”. Subsequently, STRING and VENN analyses identified a key gene group within the model, and further bioinformatics analysis revealed three hub genes in the module. Furthermore, FDX1 expression is closely linked to immune infiltration and predicts a favorable response to the immune checkpoint blockade within the tumor microenvironment. Genetic mutations and deletions increase the likelihood of antitumor immunity.

To investigate the conserved oncogenic role of FDX1 in various tumors, in vitro experiments were conducted, including knockdown of FDX1 using techniques such as RT-qPCR, Western blotting (WB), and flow cytometry (FCM). In order to provide a comprehensive analysis across multiple cancers, our manuscript was designed and analyzed according to the flow chart presented in Figure 1.

### 2.2. Landscape of FDX1 Expression across Healthy and Cancer Tissues

We explored the different expressions of FDX1 in various types of cancers from the TIMER2.0 of TCGA databases. As shown in Figure 2A, the expression levels of FDX1 in tumor tissues were distinctly lower than in healthy tissues, including BRCA, CHOL, COAD, GBM, KICH, KIRC, KIRP, LUAD, LUSC, and THCA (*p* < 0.001). However, FDX1 expression was shown to be higher in cancer samples compared with healthy samples in STAD (*p* < 0.001). In addition, we also obtained box plots of FDX1 expression in different types of cancer across the R package. As shown in Figure 2B, some types of cancers such as KICH, LIHC, THYM, DLBC, COAD, LUAD, READ, STAD, BLCA, PRAD, LUSC, PAAD, GBM, KIRC, and ESCA maintained higher expression levels of FDX1 than their control samples; FDX1 levels were particularly high in ACC. Moreover, Figure 2C shows high levels of FDX1 expression in STAD and SKCM samples via the TCGA and GTEx datasets. Therefore, the level of FDX1 expression is relatively differentially expressed in healthy and tumor tissues, making it an ideal target for tumor-specific silencing.

### 2.3. Significant Relationship between FDX1 Expression and Distinct Cancers in Clinical Characteristics

Our first finding was that patients’ age may be a vital factor in FDX1 expression, and we compared the expression of FDX1 in 33 types of cancer and found that the expression of FDX1 across various types of cancers differed according to the patients’ age. CHOL, KICH, LAML, MESO, OV, PCPG, and SKCM showed lower levels of FDX1 gene expression when patients were younger than 65 (Figure 2D). In particular, the FDX1 contents of OV and SKCM were lower than those in other tumors (Figure 2D). Conversely, people over 65 years of age with tumors such as ACC, ESCA, LUAD, KIRP, LGG, LIHC, TGCT, UCEC, UCS, and UVM retained high levels of FDX1 gene expression. Similar to the age condition, FDX1 was expressed highly in female patients with ACC, BRCA, CESC, CHOL, ESCA, HNSC, KICH, OV, UCEC, and UCS, but the opposite trend was observed for males, who expressed low levels of FDX1 gene expression in DLBC, LAML, LIHC, PRAD, READ, and TGCT (Figure 2E). In contrast, as patients became older, particularly those over 65 years old, tumors such as ACC, ESCA, LUAD, KIRP, LGG, LIHC, TGCT, UCEC, UCS, and UVM maintained high levels of FDX1 gene expression; the FDX1 contents of ESCA, LUAD, and UCEC were higher than the others. Similar to the age condition, FDX1 was expressed highly in female patients in many types of cancers, including ACC, BRCA, CESC, CHOL, ESCA, HNSC, KICH, OV, UCEC, and UCS, but we found the opposite correlation in males, who showed low gene expression in DLBC, LAML, LIHC, PRAD, READ, and TGCT (Figure 2E). As shown in Figure 2F, FDX1 expression increased at certain stages of various types of cancer. ACC and ESCA, for example, expressed high levels of FDX1. Other types of tumors such as KICH, LICH, and THCA decreased in the last three stages. According to the findings, clinical characteristics such as age, gender, and clinical stage of disease are strongly associated with the level of FDX1 expression. Furthermore, we discovered that FDX1 was highly associated with the clinical pathology stage of LIHC and THCA (Figure 2G). As shown in Figure 2F, the expression of FDX1 kept increasing in certain stages across various types of cancer. For example, FDX1 was expressed highly in ACC and ESCA. Expression in other types of tumors such as KICH, LICH, and THCA decreased in the last three stages.

Our prognostic and predictive values of FDX1 alterations in multiple types of cancer are shown in Figure 3A–D. The expression of FDX1 was associated with prognosis in patients with multiple types of cancer. In particular, FDX1 levels were significantly related to lower risk in CESC (HR = 0.617) and KIRC (HR = 0.452) but with high risk in LGG (HR = 2.646). Additionally, FDX1 was closely related to a decreased chance of disease-free survival (DFS) both in LIHC and THCA. The results regarding disease-specific survival (DSS) indicated that high FDX1 expression was linked to low risk in KIRC (HR = 0.352) and KIRP (HR = 0.419), and it was linked to higher risk in LGG (HR = 2.849). The results regarding progression-free survival (PFS) showed that a high expression of FDX1 was significantly associated with low risk in KIRC (HR = 0.444), MESO (HR = 0.425), and THCA (HR = 0.205) but with higher risk in ACC (HR = 1.277) and LGG (HR = 2.883) (all *p* < 0.05). As shown in Figure 3E–G, the Kaplan–Meier (KM) plotter data illustrated that a high expression of FDX1 was significantly associated with better prognosis for KIRC in terms of OS, DSS, and PFS, but they showed much worse prognosis for LGG. An increased FDX1 level was closely related to a longer survival prognosis in terms of DFS in LIHC and THCA. However, high FDX1 levels were significantly correlated with a better PFS prognosis in THCA, but they were negatively correlated in ACC (Figure 3H). Different conclusions could be drawn in multicancer cases based on FDX1 expression and prognostic survival.

### 2.4. FDX1 Expression Is a Key Association with Protein Phosphorylation and DNA Methylation in Multiple Cancers

A significant difference was observed between the FDX1 phosphorylation levels of normal and primary tumors in the CPTAC database. As shown in Appendix A, FDX1 is phosphorylated at a variety of sites, and there are significant differences between them. A noteworthy finding was that the data for lung cancer and hepatocellular carcinoma did not differ significantly from the data for 12 other types of human tumors (Appendix A). A high phosphorylation status was found at the S177 locus in lung cancer cases compared to healthy cases. In contrast, hepatocellular carcinoma presented completely opposite results in both S159 and S177. The PhosphoNET method was also used to assess the PDX1 phosphorylation identified by CPTAC. The FDX1 phosphorylation at the S177 locus in hepatocellular cancer functioned in the role of cellular signal transduction, and some publications supported this result. S177 phosphorylation plays a crucial role in tumorigenesis, and further study is warranted. In addition, we analyzed the methylation levels of FDX1 in healthy samples and primary samples using the TCGA, evaluating the significant differences in the FDX1 methylation levels of twenty-four different human tumors, including CHOL, UCEC, SARC, TGCT, THCA, AAD, PRAD, PCPG, READ, KIRP, LIHC, ESCA, KIRC, and COAD (Appendix A). Additionally, we examined the correlation between FDX1 methylation and STAD and SKCM pathogenesis based on the MEXPRESS database. FDX1 expression at diverse probes in the non-promoter region showed a significantly negative association with DNA methylation in both STAD and SKCM (Appendix A).

### 2.5. Genetic Alteration in FDX1 Is Closely Associated with TMB/MSI

Mutations in the FDX1 gene were examined in all cancer types to determine whether they were associated with a poor clinical outcome. As shown in Figure 4A, the highest alteration frequency in FDX1, including mutation, amplification, and deep deletion, existed in UCS cancer patients. The copy number deletion of FDX1 as the main type exists in TGCT tumors with an alteration frequency of >3%. The “amplification” mutation type almost presented in KICH (>2%), LAML (>1%), and LGG. Figure 4B shows the genetic alteration types, sites, and case numbers of FDX1 and the missense (num 17) as the primary types. R149Q in Fer2 studied in UCEC tumors might result in a frame shift mutation in FDX1, resulting in the translation of arginine (R) to glutaurine (Q) at the 149 sites, which would then lead to the truncation of the FDX1 protein. A 3D structure of the FDX1 protein with R149Q sites was presented. Moreover, we observed the possible correlation between FDX1 mutation and the clinical prognostic survival of different cancer patients. As shown in Figure 4C, FDX1-affected breast cancer patients had a better overall survival prognosis (*p* = 0.045) from 40 to 70 months after diagnosis. However, lung cancer patients with an altered FDX1 gene had worse overall survival (*p* = 5.543 × 10^−3^). Furthermore, we examined the association between FDX1 variation and TMB or MSI in all cancer types. As shown in Figure 4D,E, the expression of FDX1 with TMB was positively related to UCEC, STAD, PRAD, LGG, HNSC, and ESCA but negatively related to THYM, THCA, LUAD, KIRC, and KICH. The expression of FDX1 was also positively correlated with MSI in UCEC, STAD, KIRC, HNSC, and DLBC but negatively correlated with ACC, PAAD, LUSC, and LUAD (all *p* < 0.05).

### 2.6. The Influence of FDX1 between Prognostic Survival and Anticancer Immunotherapy Efficacy

The alteration of FDX1 could play a significant role in immunotherapy against cancer. We observed negative associations between FDX1 expression and immune cells within the TME in ACC and THCA, while positive associations were observed with PCPG, SARC, and LGG (Figure 5A,B). High FDX1 expression differed in immune and molecular subtypes within multiple human cancers and was highly expressed in the LIHC, STAD, KIRC, and PRAD of the immune subtypes. We also found high FDX1 expression in the C4 (lymphocyte depleted) and C5 (immunologically quiet) of ACC, PCPG, THCA, LGG, and SARC, and other cancers were shown based on the TISIDB dataset (Appendix A). 

FDX1 was significantly negatively expressed in the cancers ACC and THCA in the immune score, but we found it to have a positive correlation with PCPG, SARC, and LGG (Figure 5A,B). We also found FDX1 was differently negatively expressed in the cancers ACC and STAD but positively expressed in LGG and TGCT in the stromal score.

In addition, we found a negative relationship between FDX1 and stromal cells within the TME in both ACC and STAD but a positive relationship between LGG and TGCT (Figure 5C). Figure 5D,E indicate that FDX1 was negatively expressed in the CD8^+^ T cells in ACC, the M0 macrophage cells in KICH, and the monocyte cells in UVM. However, we found it to be positively expressed in the eosinophils in ACC and in the monocytes in KICH. We also clarified elevated FDX1 levels in multi-immune cells, including T cell follicular helpers, monocytes, M0 macrophages, and resting mast cells (Figure 5F). In a study of FDX1 levels and cancer immune dysfunction and exclusion (TIDE), we discovered that FDX1 levels significantly contribute to cancer immune escape (Figure 5G). We also found high FDX1 expression in the C2 (IFN-γ-dominant) of SKCM, LGG, UCEC, UCS, and PAAD and in the C4 (lymphocyte-depleted) of BRCA, CESC, CHOL, BLCA, ESCA, LIHC, HNSC, KICH, KIRC, KIRP, LUAD, THCA, UVM, LUSC, MESO, OV, TGCT, SARC, PRAD, and GBM.

In Figure 5H, we illustrate that high FDX1 levels were differently positively expressed on the class I MHC and type II IFN-response immune functions. To analyze the association between the FDX1 expression and anticancer immunotherapy, we studied a gene expression profile for the characterization of the PD-L1 blockade in a melanoma model from the Gene Expression Omnibus (GEO) of the NCBI dataset, and the results negatively correlated high FDX1 expression with immune responses through the GSE172320; however, they showed that increased FDX1 levels were positively related to immune responses via GSE22155 (Figure 5I).

### 2.7. The Analysis of Antitumor Immunotherapy in Combination FDX1 Expression with ICPs

To explore the co-expression association between FDX1 level and ICPs, we obtained results that showed a close positive association between FDX1 and PD-L1 (CD274) for BRCA, KICH, KIRC, KIRP, LGG, SARC, SKCM, TGCT, UCEC, and UVM but showed a negative association in ACC, LAML, LIHC, LUAD, STAD, and THCA (Figure 5J). Furthermore, in vitro experiments were conducted to validate the relationship between FDX1 expression and PD-L1 expression in SKCM cancer using melanoma cells from WM115 and A375. SKCM cancer showed high FDX1 expression via Western blotting (Beyotime, Shanghai, China) and RT-qPCR (Novoprotein, Shanghai, China) (Figure 6A,B). Our experimental data showed that PD-L1 protein expression appreciably decreased in WM115 and A375 cells by FDX1 knockdown based on flow cytometry (Figure 6C). Subsequently, we further detected FDX1 expression in healthy and tumor tissues by immunohistochemistry based on the HPA database (Figure 6D). All above experiments were consistent with the bioinformatic analysis.

FDX1 is especially expressed in SKCM, immune, and non-immune cells of stomach cancer cells, as shown by the scRNA-seq data from the HPA database. There are ten main cell types (T cells, Langerhans cells, fibroblasts, endothelial cells, basal keratinocytes, suprabasal keratinocytes, smooth muscle cells, and melanocytes), and fibroblast cells were separated into four subclusters (c-1, c-2, c-3, c-9). Langerhans cells were separated into two subclusters (c-0, c-12), with a different color representing the expression of FDX1 (Figure 6E).

### 2.8. The Analysis of GO/KEGG and Gene Set Enrichment of FDX1

The STRING dataset was used to obtain 100 FDX1-binding protein experiments, and the TCGA dataset was used to obtain 48 FDX1-correlated genes. After that, we studied the relationship between FDX1 and public genes and identified three hub genes related to FDX1, namely CYP11A1, FDXR, and POR, using the VEEN tool and PPI networks (Appendix A). We utilized the STRING dataset to obtain the 100 available FDX1-binding protein experiments, as well as the first 48 FDX1-correlated genes via the TCGA dataset. Then, we continued to analyze the relationship between FDX1 and public genes and attained three hub genes, namely CYP11A1, FDXR, and POR, which are closely related to FDX1, and visualized them with the VEEN tool and PPI networks with the TIMER2.0 tool. A positive relationship was found between FDX1 expression and CYP11A1 (R = 0.69), FDXR (R = 0.73), and POR (R = 0.43) (Appendix A). Furthermore, we investigated the functions of FDX1-related genes using GO functional annotation and KEGG pathways (Figure 6F,G and Appendix A). The results indicated that “pathways in cancer”, “melanogenesis”, “metabolic pathway process”, “cellular response to oxidative stress”, “mitochondrion”, “G-protein coupled receptor signaling pathway”, and “chemokine signaling pathway” may have a potentially vital impact on the clinical pathogenesis of multiple types of tumors (Appendix A).

We also identified relevant gene set pathways of FDX1 through the GSEA tool in multiple types of cancer. FDX1 expression was found to have a close relationship with the immune- and matrix-related pathways in UVM, THCA, THYM, STAD, SARC, PCPG, PRAD, READ, OV, LUSC, ACC, LGG, LAML, KIRC, COAD, BLCA, and LUAD, including T cell receptors, nod-like receptors, and the chemotaxis factor pathways. In addition, FDX1 was enriched in oxidative and metabolism pathways, as well as others, in UCS, HNSC, ESCA, and CHOL (Appendix A). In general, FDX1 expression was significantly associated with many essential pathways in human cancer formation.

### 2.9. FDX1 Gene Expression Is Positively Correlated with Immune Infiltration in Tumor Immune Microenvironment

In the analysis conducted using the TCGA database, a negative correlation was observed between FDX1 levels and cancer-associated fibroblasts across multiple cancer types. Specifically, FDX1 levels were significantly lower in ACC, COAD, ESCA, KIRC, KIRP, STAD, THCA, and UCEC. However, the positive expression of FDX1 was detected using four algorithms (EPIC, MCPCOUNTER, XCELL, and TIDE) in BRCA-lumb, HNSC, HNSC-HPV-, LGG, LIHC, PCPG, SKCM, and TGCT (Figure 7A). Furthermore, the relationship between FDX1 expression and CD8+ T cell infiltration was investigated. According to seven algorithms (TIMER, EPIC, MCPCOUNTER, CIBERSORT, CIBERSORT-ABS, QUANTISEQ, and XCELL), FDX1 was positively expressed in BLCA, BRCA-lumb, CESC, ESCA, KICH, THYM, and UVM. Conversely, FDX1 expression was found to be negative in ACC, HNSC, HNSC-HPV+, and STAD. Notably, CIBERSORT analysis revealed differential expression of CD8^+^ T cell infiltrations in SKCM, which could serve as a basis for further investigation (Figure 7B).

### 2.10. The Association between FDX1 and DNAss/RNAss and Drug Sensitivity Analysis

DNAss and RNAss were used to validate the relationship between FDX1 level and cancer stemness. Then, the data of DNA-methylated pattern and mRNA expression were provided to assess the cancer stemness. The result indicated that the FDX1 level significantly increased in cancer-initiating stem cells, and it functions in cancer resistance. For example, high FDX1 expression in cancer stem cells decreased proliferation and overturned the epithelial–mesenchymal transition via tumor resistance to ammonified in human melanoma and promoted the treatment of resistant acute myeloid leukemia. Interestingly, FDX1 expression was not always the same in both DNAss and RNAss. As shown in Figure 8A, there was a significantly positive correlation between FDX1 level and DNAss in COAD, LGG, OV, PCPG, SARC, SKCM, and STAD, but there existed a negative correlation in KICH, LAML, TGCT, THCA, and THYM. We also found a positive relationship between FDX1 and RNAss in the cancers ACC, COAD, ESCA, KIRC, KIRP, OV, PAAD, PRAD, SARC, SKCM, STAD, THCA, THYM, UCEC, UCS, and VUM but completely opposite results in CHOL, LGG, and TGCT. Here, the results of SKCM and STAD are visualized by a scatter plot. These clearly opposite results indicate that the function of DNAss and RNAss might clarify diverse tumorous cell population characteristics based on cancer stemness features in multiple cancer types.

The tumor stemness analysis demonstrated the correlation between FDX1 and the DNAss/RNAss value; we also found a positive relationship between FDX1 and SKCM and STAD. A higher DNAss/RNAss value was linked to adverse prognosis, tumor mutational burden, and immune and stromal cells, which may indicate the benefit of antitumor immunotherapy and the reliability of survival prognosis in terms of the significance of DNAss/RNAss value (Figure 8B).

Further investigations were conducted on the association between FDX1 level and cancerous stem-cell-like characteristics regarding the drug sensitivity of 200 chemotherapy drugs (assessed by Z-score; higher score denotes more sensitive to treatment) based on NCI-60 human tumor cell lines (all *p* < 0.05). The results demonstrate that high FDX1 expression is strongly positively associated with the upgraded drug sensitivity of several human cell lines to chemotherapeutic drugs, expect for everolimus and ammonified with increased drug resistance (Figure 8C), such as resistance to ifosfamide (treatment for recurrent testicular cancer and germ cell tumors, sarcomas, non-Hodgkin’s lymphoma, Hodgkin’s disease, non-small-cell and small-cell lung cancer and bladder cancer, head and neck cancer, and cervical cancer). Moreover, we identified the association between FDX1 level and several drug sensitivities valued by clinical trial or FDA approval. The results indicated that a low FDX1 level correlated to higher sensitivity in cancers for erlotinib, salubrinal, HG-6-64-1, MS-275, bortezornib, and WH-4-023, but they showed that a high FDX1 level that indicated increased drug sensitivity (IC50) in drugs including sunitinib, thapsigargin, VX-680, PF-562271, PHA-665752, LY317615, QL-XII-47, obatoclax mesylate, PAC-1, paclitaxel, XMD14-99, Z-LLNe-CHO, JW-7-52-1, fluorouracil, ZSTK474, AT-7519, Z-LLNle-CHO, BI-2536, bleomycin, BMS345541, BMS-509744, CAL-101, XMD14-99, PAC-1, epothilone-B, FMK, FR-180204, FTI-277, gemcitabine, obatoclax mesylate, GSK-650394, GSK1070916, GSK1904529A, GW843682X, IPA-3, LY317615, JW-7-52-1, NSC-87877, bortezomib, eriotinib, PF-562271, ruxolitinib, S-TriyL-cysteine, and tipifarnib (Appendix A).

### 2.11. The Effect of FDX1 Expression on Antineoplastic Drug Sensitivity

To further identify the possible targeting role of these drugs on FDX1, we first investigated the association between chemotherapeutic drug sensitivity and FDX1 expression (Figure 8D,E). In addition, we examined the binding mode between FDX1 and different drugs using a molecular docking tool in order to identify any potential interactions between them. Chemotherapeutic drugs bind inside the pocket of FDX1 with a compact formation. An aryl group of everolimus bound to the hydrophobic pocket, surrounded by the residues ARG421, ARG357, and TRP418 to form a stable and robust hydrophobic binding. Moreover, ten vital hydrogen bonds were displayed, including erlotinib, salubrinal, defactinib, WH-4-023, everolimus, chelerythrine, elesclomol, Z-LLNLE-CHO, 5-fluorouracil, and ifosfamide, with hydrogen bonding forces of 15, 11, 11, 17, 10, 10, 10, 10, 15, and 15, respectively. These results show the potential main interaction between FDX1 and the top 10 chemotherapeutic drugs to enable multiple drugs to anchor in the diverse binding sites of FDX1 (Appendix A). All these results presented chemotherapeutic drugs with excellent biological binding activities, promoting their antitumor effects.

## 3. Discussion

Mammalian adrenodoxin is involved in and contributes to the biosynthesis of (Fe-S) clusters, which is important for the synthesis of various steroid hormones in adrenal glands [10]. A ferredoxin, especially FDX1, functions as an electron acceptor or donor and is involved in redox signaling. FDX1 plays a key role in the electron transport chain (ETC) by accepting electrons from ferredoxin-NADP^+^ oxidoreductase (FNR). An analysis of the GO enrichment of FDX1 in biological processes (BPs) focused primarily on steroid metabolism, sterol metabolism, and cholesterol metabolism. The significant functional enrichment of FDX1 associations with steroid pathways suggests that cholesterol might be lowered by sterol, resulting in FDX1 mediating the change in steroid metabolism. In addition, other studies have demonstrated that high FDX1 expression can contribute to a variety of biochemical metabolic pathways that can impact tumor development and progression. On the basis of phylogenetic trees on the HomoloGene website, the HomoloGene analysis of FDX1 with diverse species indicated that FDX1 may present similar functional mechanisms under normal physiological conditions for the conservation of the FDX1 protein structure across diverse species. Studies have reported the functional link between FDX1 and clinical disease, particularly multicancers. It is unknown whether FDX1 plays a role in the pathogenesis of multiple types of cancer across potential molecular mechanisms.

The ferredoxin family comprises the conserved iron sulfur [2Fe-2S] cluster binding motif and binding domain, in which the binding motif of vertebrate adrenodoxins are an essential component of the mitochondrial metabolism. Ferredoxins function as electron acceptors or donors and engage in redox signaling, especially FDX1. As the first stromal electron acceptor in the electron transport chain (ETC), FDX1 plays a key role in transferring electrons to ferredoxin-NADP+ oxidoreductase (FNR) or other diverse proteins. The ferredoxins exist in different species, including algae, bacteria, and higher plants and animals, and consist of soluble iron sulfur [Fe-S] cluster proteins. Importantly, they operate as electron donors to participate in a series of key metabolic pathways in diverse organisms. The GO enrichment analysis of FDX1 in biological processes (BPs) mainly focuses on these steroid, sterol, and cholesterol metabolisms as well as steroid biosynthesis. The significantly functional enrichment of FDX1 associations maps to the steroid pathway; a hypothesis is that sterol might lower cholesterol levels, resulting in FDX1 mediating the change in steroid metabolism. FDX1 consists of Fe-S clusters with a redox active group, playing an essential role in the biosynthesis of diverse steroid hormones in adrenal glands and functioning to reduce cytochrome P450 in the mitochondria to promote the conversion of cholesterol to pregnenolone and cortisol. Pregnenolone acts as an active neurosteroid to protect and strengthen cognition and antidepressant effects in the brain. Therefore, high FDX1 expression might promote a variety of biochemical metabolic pathways, further impacting tumorous development and progression. Studies have reported that multifunctional FDX1 protein is involved in distinct cellular biological processes, including the regulation of metabolism and oxidoreductase activity, and the cellular response to cAMP and the P450-containing electron transport chain. The HomoloGene analysis of FDX1 with diverse species based on the phylogenetic tree and HomoloGene websites displayed the conservation of the FDX1 protein structure, which indicated that FDX1 may present similar functional mechanisms in healthy physiological conditions.

Based on potential molecular mechanisms, it is unknown whether FDX1 plays a role in the pathogenesis of multiple types of cancer. A comprehensive and systematic analysis of the FDX1 gene in thirty-three distinct multiple cancer types was conducted using the TCGA, GEO, CAPTAC databases and cBioPortal website [50,51], in addition to molecular functions such as gene expression, somatic mutations, methylated DNA, and protein phosphorylation analysis. According to the TCGA and GTEx datasets, FDX1 expression differs significantly between multiple cancer types, which indicates that FDX1 level is closely related to patients’ clinical prognosis. In KIRC, the presence of higher levels of FDX1 was closely correlated with better prognoses for OS, DSS, and PFS, but we also found that LGG tumors had poorer prognoses. Based on Kaplan–Meier plots and Cox regression survival analysis, an increased FDX1 level significantly predicted good overall survival in LIHC, THCA, and THCA, whereas a lower FDX1 level predicted better overall survival in ACC. Additionally, we found that high FDX1 expression was strongly associated with an age over 65, especially in ESCA, LUAD, OV, SKCM, and UCEC cancers. A high level of FDX1 is expressed mainly in females in both BRCA and KIRC. The data also demonstrated that the level of FDX1 was closely correlated with distinct pathological stages in patients with multiple cancer types. FDX1 expression was significantly associated with a better prognosis for survival in ESCA, KICH, LIHC, and THCA cancers. Based on the results above, it appears that the expression of FDX1 has a significant clinical prognostic value in certain types of cancer. The molecular mechanisms involved in the pathogenesis of multiple cancers may be similar to those involved in FDX1.

Accumulating evidence suggests that tumor-infiltrating immune cells in the tumor microenvironment (TME) play a crucial role in cancer prevention. The results of the analysis demonstrate a strong association between FDX1 expression and immune cell infiltration in the TME across multiple cancer types. Specifically, a significant negative correlation was observed between FDX1 levels and ACC levels in terms of stromal and immune scores, as well as CD8^+^ T cells. Conversely, a positive correlation was found between FDX1 levels and eosinophil levels. High FDX1 levels in both the stroma and immune scores were closely associated with LGG. In KICH, a high level of FDX1 on monocytes showed a significant positive association with higher FDX1 levels. On the other hand, FDX1 levels on M0 macrophages and monocytes in UWM displayed a negative relationship. Furthermore, low levels of FDX1 were closely associated with resting mast cells compared to controls with normal levels. The adjacent infiltrated stromal fibroblasts were activated and transdifferentiated into cancer-associated fibroblasts, stimulated by constant paracrine cell-to-cell communication. Based on these findings, it can be concluded that high levels of FDX1, associated with greater immunogenicity of the TME, play a key role in the effectiveness of immune checkpoint therapies and CD8^+^ T cell infiltration against tumors.

According to reports, tumor cells can construct complex TMEs where several immune cells contribute to the invasion, development, and progression of cancer, as well as resistance to immunotherapy, which allows tumor cells to escape the immune response. In our study, we found that genetic mutations and deletions in FDX1 might decrease MHC class I (MHC_I) levels, which affects how well peptides bind to MHC molecules and how well endogenous antigens are presented to CD8^+^ T cells, resulting in cytotoxic T lymphocytes failing to recognize and kill tumor cells. The TIDE tool showed that increased FDX1 levels could lead to higher immune escape and a low response rate to anticancer immunotherapy. Furthermore, we investigated the highest frequency of somatic mutations both at high and low FDX1 levels as a predictive biomarker assessed by the TMB and MSI approaches, in which tumors produced abnormal proteins combined with MHC molecules, which were recognized as non-self by effector T cells for elimination. PD-L1 was reported to be associated with PD-1 in the TME, which contributes to the activation of immunosuppressive functions. As we know, IFNs are critical in antitumor immune responses, which causes PD-L1. Additionally, it was shown that high levels of FDX1 contribute to promoting IFN responses, that cytokines could be used to enhance antitumor immunotherapy, and that adaptive immune responses could result in long-term immunity. Therefore, we conducted an external validation with GSE172320 and GSE22155 in patients with melanoma who responded to a PD-L1 blockade. The results showed that melanoma patients with higher FDX1 expression might present with different efficacy of PD-L1 immune response during diverse immunotherapy stages.

A further concern is the influence of FDX1 expression on antineoplastic drug resistance. Based on our molecular docking analysis, we found that 10 of the top 11 drugs directly interact with FDX1. An extensive study indicated that more sensitive drugs could target FDX1 as well as show significant biological binding activity towards FDX1. Whether FDX1 could function in the pathogenesis of distinct multiple cancer types based on potential molecular mechanisms has not been well described. In our study, we comprehensively and systematically analyzed FDX1 in thirty-three distinct cancer types via the TCGA, GEO, and CAPTAC datasets, as well as the molecular functions of gene expression, somatic mutations, methylated DNA, and phosphorylated proteins.

As we know, a functional association exists between FDX1 and clinical disease, particularly in diverse tumors [20,52,53,54,55,56,57,58]. FDX1 expression was found to be significantly different in multiple cancer types based on the TCGA and GTEx datasets, indicating that FDX1 level is remarkably associated with the clinical prognosis of patients with multiple cancer types. High FDX1 levels were closely correlated with better prognosis in terms of OS, DSS, and PFS in KIRC, but we also found a worse prognosis for LGG tumors. Additionally, an increased FDX1 level was closely linked to excellent DFS prognostic outcomes in the cancers LIHC, THCA, and THCA. In contrast, lower FDX1 levels were closely related to better DFS prognosis in ACC based on Kaplan–Meier plots and the Cox regression survival approach. Furthermore, we observed that high FDX1 expression was strongly associated with age greater than 65, especially in the cancers ESCA, LUAD, OV, SKCM, and UCEC. Additionally, high FDX1 levels were mainly expressed in females in both BRCA and KIRC. Additionally, the data demonstrated that FDX1 level was closely related to distinct pathological stages of patients with multiple cancer types; in particular, FDX1 expression indicated a significantly better prognostic survival rate in cancers ESCA, KICH, LIHC, and THCA based on the hepia2 tool and R package “survival”. All above results illustrated that FDX1 expression might act as an excellent clinical prognostic value for patients with certain cancers. Additionally, FDX1 might function in the pathogenesis of multiple cancer types via similar molecular mechanisms.

Growing evidence has illustrated that tumor infiltration immune cells of the TME work in a key role against tumors. The results displayed that FDX1 expression is strongly associated with immune cells’ infiltration of the TME in multiple cancer types. The FDX1 level is significantly negatively correlated with ACC in stromal and immune score, as well as CD8^+^ T cells, but we found that it is positively related to eosinophils. A high FDX1 level is closely positively linked with LGG both in stromal and immune score. Moreover, a high FDX1 level was significantly positively related to monocytes in KICH but negatively to M0 macrophages as well as monocytes in UWM. Additionally, a low FDX1 level was closely associated with resting mast cells than in the control. The adjacent infiltrated stromal fibroblasts were activated and transdifferentiated into cancer-associated fibroblasts stimulated by constant paracrine cell-to-cell communication. These results imply that high FDX1 expression, with greater immunogenicity of the TME to promote CD8^+^ T cells’ infiltration and the efficacy of immune checkpoint therapy, plays a key role in antitumor immunotherapy.

Studies have reported that tumor cells construct a complex TME, where several immune cells increase cancer invasion, development, and progression and resistance to immunotherapy, leading to the escape of tumor cells from immune response. We found that genetic mutation and deletion in FDX1 might decrease the level of MHC class I (MHC_I) so that it fails to efficiently bind peptides to MHC molecules and presents the endogenous antigen to CD8^+^ T cells, causing cytotoxic T lymphocytes to fail to recognize and kill tumor cells. We also found that an increased FDX1 level might cause a higher immune escape with a low response rate to anticancer immunotherapy based on the TIDE tool. Furthermore, we explored the highest mutation frequency of somatic mutations both in high and low FDX1 levels as a predictive biomarker assessed by the TMB and MSI approaches, in which tumors generated abnormal proteins combined with MHC molecules and recognized as non-self by effector T cells to kill and remove. It has been reported that PD-L1 combines with PD-1 in the TME, which contributes to systematically activating immunosuppressive functions. As we know, IFNs are considered critical, and PD-L1 is caused by IFN γ in the antitumor immune response. We also identified that high FDX1 levels contribute to promoting the IFN γ response, and the effector cytokines can enhance the antitumor immunotherapy; then, adaptive immune responses can construct long-term and continued immunity. Thus, we conducted an external validation experiment with both GSE172320 and GSE22155 in melanoma patients’ immune response to PD-L1 blockade. The result indicated that increased FDX1 expression with a greater PD-L1 immune response in GSE22155 resulted in a better immunotherapy efficacy in patients, but we found the completely reverse outcome for GSE172320.

Additionally, the effect of FDX1 expression on antineoplastic drug resistance drew our attention. Our molecular docking analysis displayed that the top 10 drugs directly interacted with FDX1. Detailed studies implied the possible targeting effect of excellent drug-sensitivity drugs on FDX1 and significant biological binding activity to FDX1. The study was further divided into clinical patients with different sensitivities to antitumor drugs based on the diverse drug sensitivities of a wide array of tumorous diseases. Our results indicated that high FDX1 levels were associated with lower IC50 values in several chemotherapy drugs, including erlotiib, salubnrinal, HG-6-64-1, MS-275, bortezornib, and WH-4-023, which were considered better choices for tumorous patients. For patients with low FDX1 levels, fluorouracil, ZSTK474, AT-7519, Z-LLNle-CHO, BI-2536, and bleomycin might be better options than drugs with a high IC50.

In summary, FDX1 could serve as a novel antitumor immunotherapy biomarker and predictor. FDX1 regulation with a combination of immuno-checkpoint inhibitor therapy and chemotherapy drugs to promote the antitumor immunotherapy of multiform cancers will contribute to our development of antitumor drugs in the future. In view of the diverse drug sensitivities in a wide variety of tumorous diseases in patients, we further divided clinical patients with different sensitivities to antitumor drugs into treatment groups, and the results implied that for high FDX1 levels, chemotherapy drugs with lower IC50 values, including erlotiib, salubnrinal, HG-6-64-1, MS-275, bortezornib, and WH-4-023, are excellent choices for tumorous patients. However, other drugs such as fluorouracil, ZSTK474, AT-7519, Z-LLNle-CHO, BI-2536, and bleomycin, with high IC50 values, might be better in patients in low FDX1 levels.

It is hypothesized that regulating FDX1 using a combination of immune checkpoint inhibitors and chemotherapy drugs can contribute to the development of antitumor drugs for various types of cancers. Therefore, FDX1 has the potential to serve as a novel clinical prognostic biomarker and predictor for antitumor immunotherapy.

## 4. Materials and Methods

### 4.1. Data Acquisition and Analysis of FDX1 Expression in Multitumors

We searched for FDX1 in the “Gene_DE” module of TIMER2 (tumor immune estimation resource, version 2) web (http://timer.cistrome.org/ (accessed on 3 September 2022)), and the expression difference of FDX1 between tumor and adjacent normal tissues for the different tumors or specific tumor subtypes of the TCGA project was observed as well as for several cancers without normal or with highly limited normal tissues (e.g., TCGA–glioblastoma multiforme (GBM) and TCGA–LAML data), Additionally, we obtained violin plots of the FDX1 expression in different pathological stages (stage I, stage II, stage III, and stage IV) of all TCGA tumors via the “Pathological Stage Plot” module of HEPIA2. The log2 [transcripts per million (TPM) + 1] transformed expression data were applied for the box or violin plots. The DNA/RNA stemness score (DNAss/RNAss) were assessed by copy number variation data through XENA of the TCGA dataset (http://xena.ucsc.edu/ (accessed on 3 September 2022)).

### 4.2. Correlation between FDX1 Expression and Diverse Cancers in the Clinical Survival Characteristics

The “Survival Map” module of GEPIA2 was used to acquire the overall survival (OS) and disease-free survival (DFS) significance map data of FDX1 across all of TCGA tumors. High (50%) and low (50%) cutoff values were used as the expression thresholds for splitting the high-expression and low-expression cohorts. The log-rank test was used in the hypothesis test, and the survival plots were also obtained through the “Survival Analysis” module of GEPIA2.

### 4.3. The Analysis of Genetic Alteration

When logging into the cBioPortal website (https://www.cbioportal.org/ (accessed on 6 September 2022)), the “TCGA Pan Cancer Atlas Studies” was chosen in the “Quick select” section, and “FDX1” was entered for queries of the genetic alteration characteristics of FDX1. The results of the nitic alteration according to Kaplan–Meier analysis are visualized as a heatmap and a scatter plot.

### 4.4. Targeting FDX1 with Molecular Docking Simulation

To explore the binding efficacy of chemotherapeutics and FDX1 protein, we first downloaded the three-dimensional structure of the FDX1 protein (PDB:3NAL) based on the PDB tool (https://www.rcsb.org/ (accessed on 6 September 2022)). Then, the ligands of the active center were dehydrated and removed through the PyMOL tool. The small molecule structures of chemotherapeutic drugs were downloaded from PubChem (https://pubchem.ncbi.nlm.nih.gov/ (accessed on 7 September 2022)). The receptor proteins were combined with small molecule ligands, such as adding polar hydrogens and charge calculations and setting up rotation bonds, based on the AutoDock Tool. Finally, the parameters of the receptor protein docking site were combined with the active pocket site of small molecule ligand binding through the AutoDock4 tool and visualized by the PyMOL approach.

### 4.5. The Expression of FDX1 in Immune and Molecular Subtypes

We accessed TISIDB (http://cis.hku.hk/TISIDB/browse.php?gene=FDX1, (accessed on 7 September 2022)) and entered the gene symbol "FDX1". We selected the "Subtype" option and chose the immune subtype. Subsequently, we retrieved the distribution of different cancer types across immune subtypes. Lastly, we focused on the molecule subtype and obtained various molecule subtypes.

### 4.6. FDX1-Related Gene Set Enrichment Analysis

The STRING website (https://string-db.org/ (accessed on 16 September 2022)) was initially utilized to investigate the protein “FDX1” in the organism “Homo sapiens”. Subsequently, a selection of genes correlated with FDX1 was chosen based on the datasets of tumor and normal tissues from the TCGA. The “correlation analysis” module of GEPIA2 was employed to perform pairwise gene Pearson correlation analysis between FDX1 and the selected genes. The dot plot utilized log2 TPM values. For annotation, visualization, and integrated discovery, the VID Dataset was employed by selecting the identifier “OFFICIAL_GENE_SYMBOL” and species “Homo sapiens”. Functional annotation chart data were obtained from this analysis. Additionally, the “clusterprofiler” R package was investigated to conduct gene set enrichment analysis (GSEA), and statistical significance was determined using adjusted *p* < 0.05.

### 4.7. Experimental Validation In Vitro Both in AGS and A375 Cells

The stomach cancer AGS cells and the melanoma A375 cells were cultured on DMEM (Gibco, Grand Island, NY, USA) with 10 percent fetal bovine serum in a 5% CO_2_ humidified incubator at 37 °C. The two cell types with about 80 percent confluence were used. AGS and A375 were grown on six-well plates and were transfected by small interference RNA (siRNA), and then after 24 h, total RNA of AGS and A375 cells were isolated by RNAiso plus agents (Takara, Kusatsu, Japan) and reverse transcribed into cDNA with GoldenstarTMRT6 cDNA synthesis kit (Tsingke Biotechnology Co., Ltd., Beijing, China), and qRT-PCR was given by SYBR qPCR Master Mix (Novoprotein, Shanghai, China). The relative primers were synthesized by RuiBo Biology (Guangzhou, China) and listed below:hGAPDH: F-GAACGGGAAGCTCACTGG; R-GCCTGCTTCACCACCTTCT;hPD-L1: F-CCAGTCACCTCTGAACATG; R-TCAGTGTGCTGGTCACATTG;hFDX1: F-GACAATATGACTGTTCGA; R-AGTTCAGGAGGTCTTGCCCA;

After 48 h, total proteins of AGS and A375 cells were extracted with RIPA lysis buffer (Applygen, Beijing, China) with added 1 percent phosphotase inhibitor cocktail (Beyotime, Shanghai, China) and 1 mM PMSF (Beijing Solarbio Science & Technology Co., Ltd., Beijing, China), and then the samples were mixed with SDS-PAEG gel. The bands were visualized by ECL chemiluminescence (Millipore, Billerica, MA, USA). After 48 h, the two cell types were collected and stained with antibodies (Biolegend, San Diego, CA, USA) for about 30 min at 4 °C. To observe PD-L1 expression on the cell surfaces, two types of tumor cell were incubated with PECY7-CD274 (PD-L1, Biolegend) for 30 min at 4 °C and then analyzed using FCM (BD Facscelesta biosciences, San Jose, CA, USA). The data were computed using the FlowJo_v10 app.

## Figures and Tables

**Figure 1 ijms-24-09182-f001:**
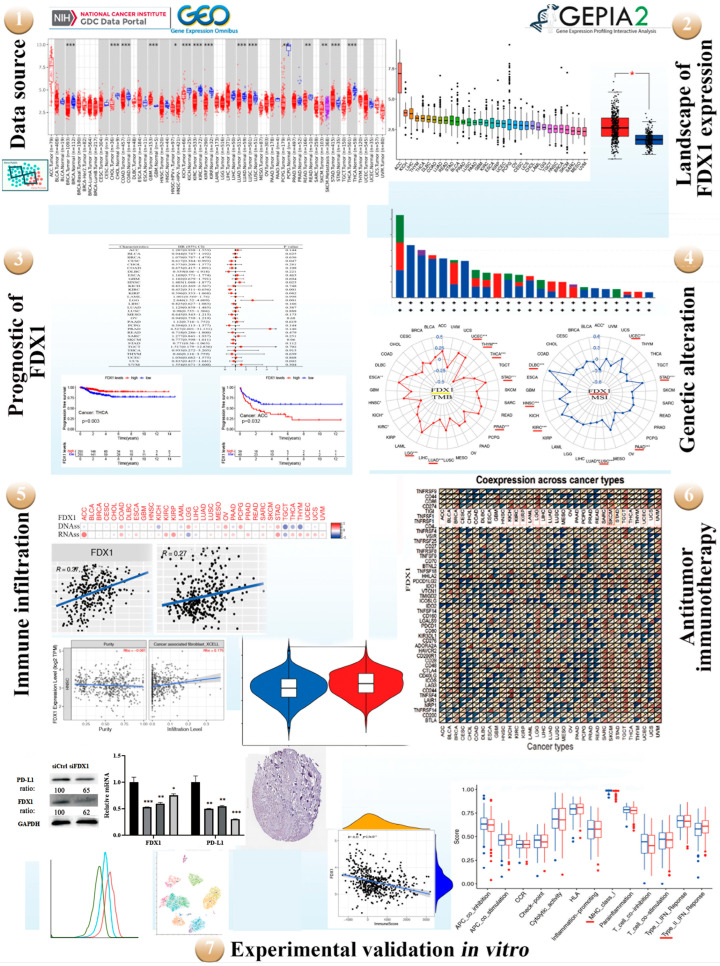
A workflow of the study. Red represents diverse tumor tissues and blue represents normal tissues. TMB and MSI divided by colors (yellow and red). Green line of flow cytometry represents isotype, blue line represents siFDX1, and red line represents siCtrl. red * *p* < 0.05, * *p* < 0.05; ** *p* < 0.01; *** *p* < 0.001.

**Figure 2 ijms-24-09182-f002:**
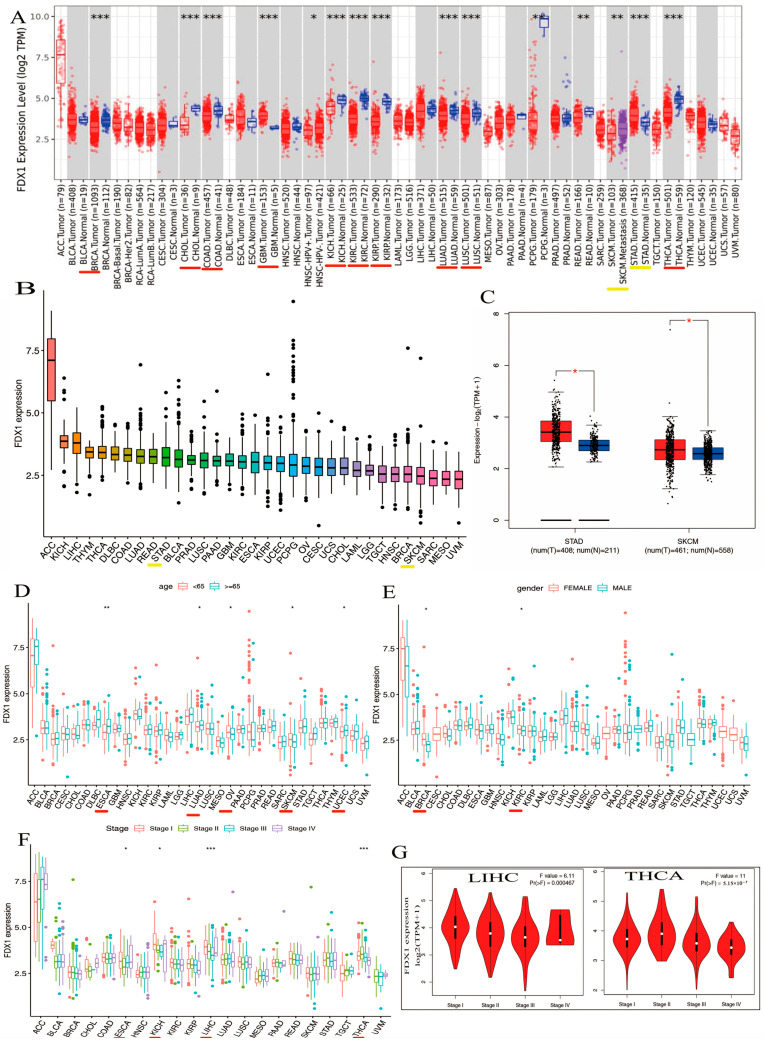
Differential expression analysis of FDX1 and clinical characteristics. (**A**) Expression levels of FDX1 in tumor and corresponding adjacent tissues in multicancer (**B**) Expression of FDX1 mRNA in pan-cancer (**C**) Expression differences of FDX1 in STAD and SKCM. Clinical correlation analysis of FDX1 in pan-cancer. (**D**) Patients’ age. (**E**) Patients’ gender. (**F**) Tumor stage. (**G**) Pathological stages in LIHC and THCA. Red underlines indicate cancers exhibiting significant differences between tumor samples and normal controls. Yellow underlines denote the two types of cancers that we will investigate in our subsequent studies. red * *p* < 0.05, * *p* < 0.05; ** *p* < 0.01; *** *p* < 0.001.

**Figure 3 ijms-24-09182-f003:**
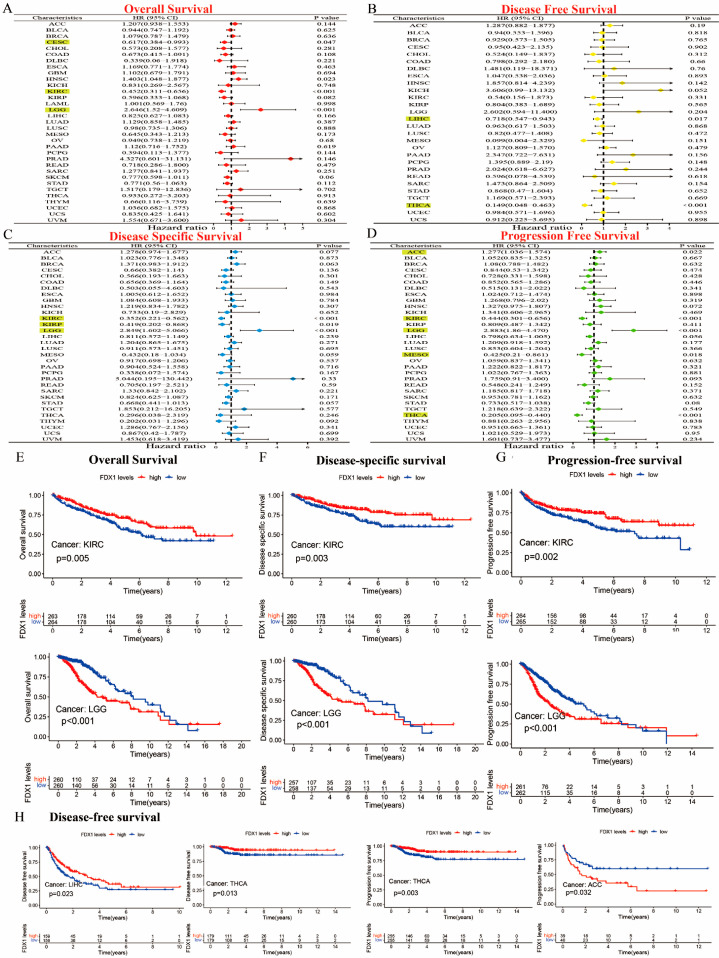
Prognostic assessment of FDX1 expression in OS, DFS, DSS, and PFS. Correlation between FDX1 expression and OS, DFS, DSS, and PFS by utilizing the forest plots of univariate Cox regression analyses. FDX1 expression was significantly correlated with prognosis in these types of cancers (*p* < 0.05) in the highlights (**A**–**D**), Kaplan–Meier analysis of OS, DFS, and PFS in patients with high and low FDX1 expression (**E**–**H**).

**Figure 4 ijms-24-09182-f004:**
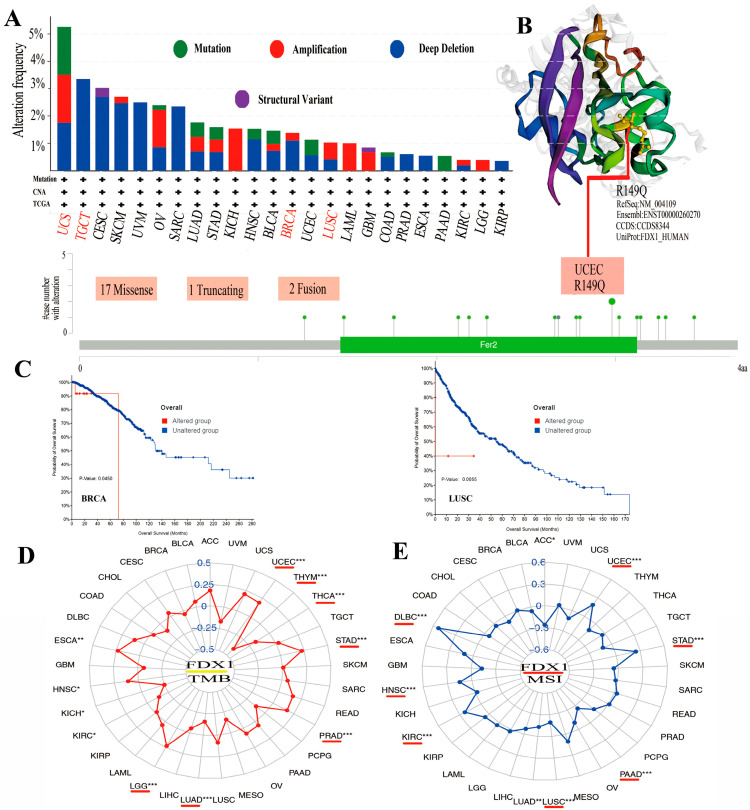
The mutation of FDX1 and TMB/MSI in human multicancers of the TCGA dataset. Through cBioPortal of the TCGA dataset, we gained the genetic mutation frequency in various tumors (**A**) and multiple mutation sites of FDX1, and the highest genetic mutation frequency was presented in the 3D structure of FDX1 (**B**). We also displayed the possible association between alteration status and overall survival of BRCA and LUSC (**C**). We found a close association between FDX1 and TMB/MSI in human multicancers of TCGA through R packages (**D**,**E**). The red lines indicate cancer types with significant correlations between FDX1 and TMB or MSI. * *p* < 0.05; ** *p* < 0.01; *** *p* < 0.001.

**Figure 5 ijms-24-09182-f005:**
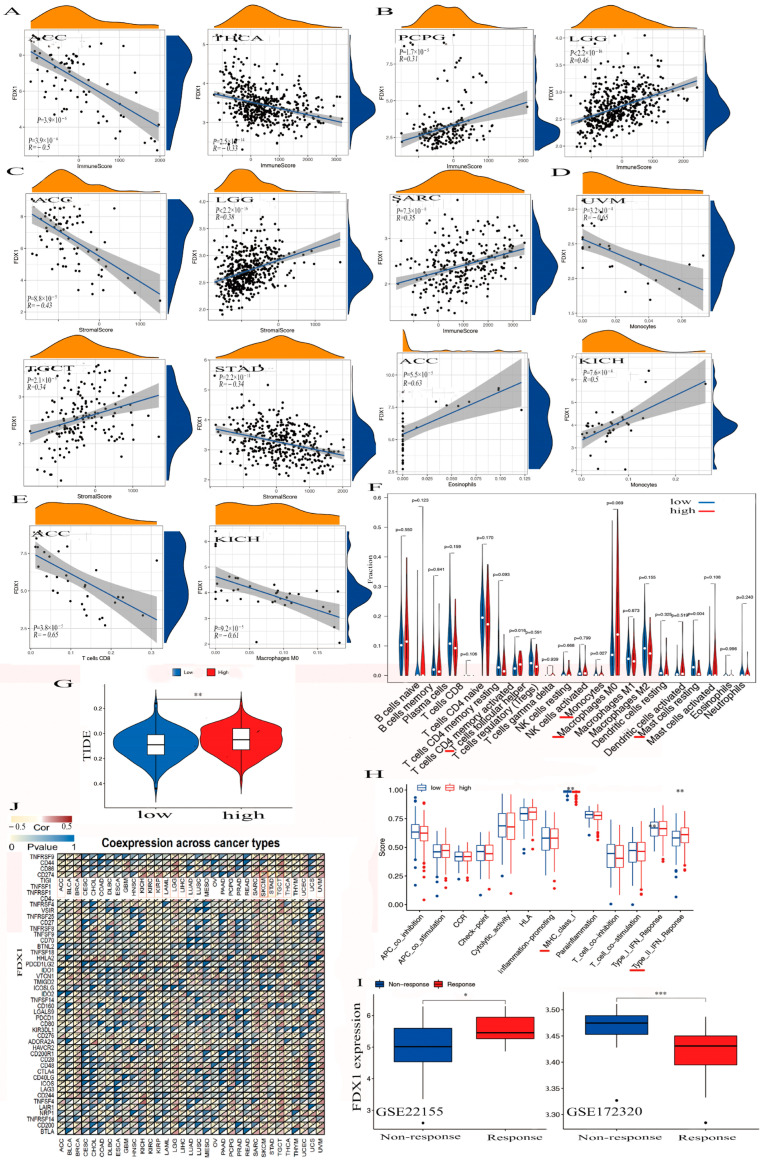
The association between FDX1 and tumor immune-infiltration cells and ICPs in human multicancers of the TCGA dataset. According to the highest difference of both immune score in ACC, THCA, PCPG, LGG, and SARC (**A**,**B**) and stromal score in ACC, LGG, TGCT, and STAD (**C**). We observed the expression of FDX1 with some immune cells in UVM, ACC, KICH, ACC, KICH, and other immune cells (**D**–**F**). We also found the correlation between high FDX1 expression and immune escape in human tumors (**G**). Further analysis of the association between FDX1 level and immune function. The red line indicates a significant difference between the high-risk and low-risk groups (**H**). Through the GEO dataset, the correlation between FDX1 and immune response to PD-L1 blockade was assessed in SKCM patients (**I**). We first explored the correlation between FDX1 expression and immune checkpoint molecules, especially PD-L1, in human multicancers (**J**). Statistically significant *p*-value calculated (Pearson correlation analysis): * *p* < 0.05, ** *p* < 0.01, *** *p* < 0.001.

**Figure 6 ijms-24-09182-f006:**
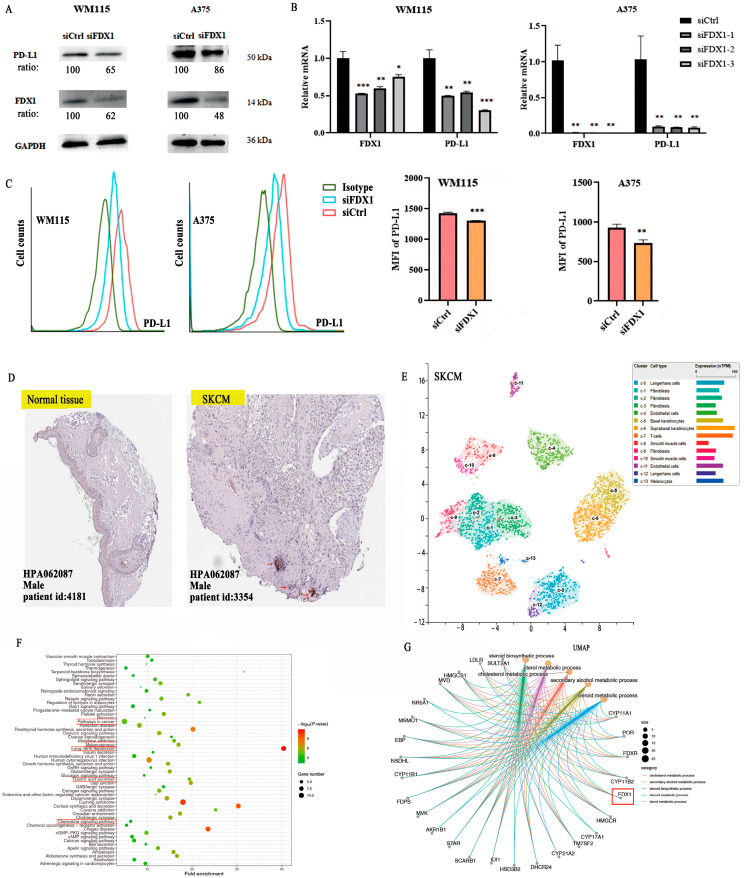
FDX1 knockdown impacts PD-L1 expression in SKCM tumors by WM115 and A375 cell lines. (**A**) Western blotting examination of the protein levels of WM115 and A375 cells. (**B**) and the mRNA level of PD-L1 of WM115 and A375 cells. (**C**) The cell counts and MFI of PD-L1 of WM115 and A375 cells examined by FCM. (**D**) FDX1 expression between tumor and adjacent normal tissues of SKCM was analyzed by immunohistochemistry based on HPA dataset (IHC staining: 100 μm). (**E**) UMAP plot depicting various cell types in SKCM cancer and divided by colors. (**F**,**G**) The enrichment pathway of KEGG and biological processes of GO of FDX1 expression. All experiments were replicated at least three times. ns: means no significant difference, * *p* < 0.05; ** *p* < 0.01; *** *p* < 0.001.

**Figure 7 ijms-24-09182-f007:**
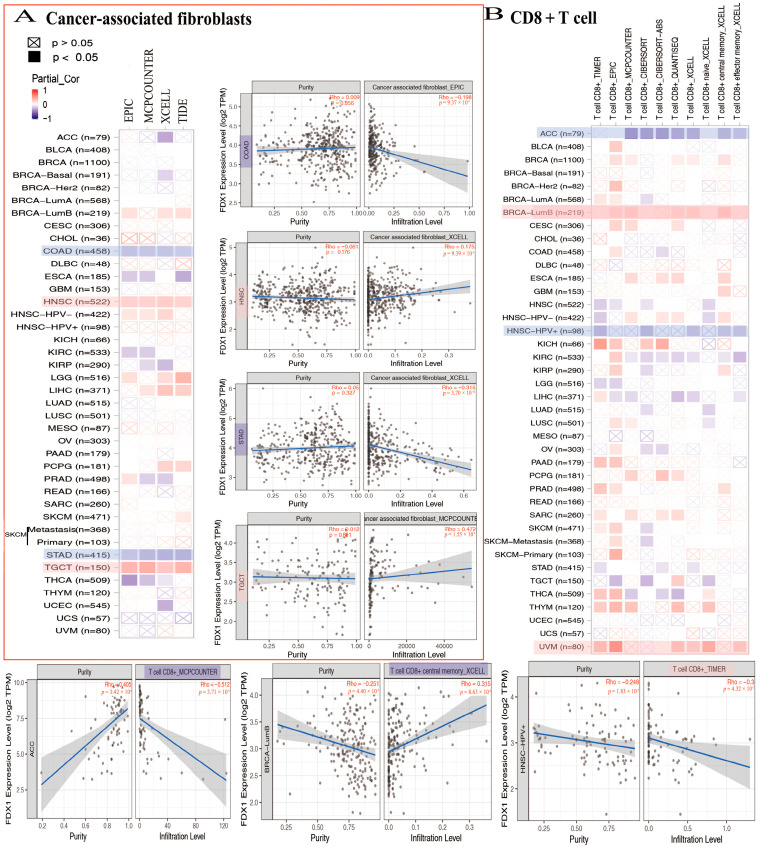
FDX1 expressed differently on the immune infiltration of cancer-associated fibroblasts and CD8^+^ T cells. We analyzed the possible correlation between FDX1 and immune-infiltrated level of CAF (**A**) and CD8^+^ T cells (**B**).

**Figure 8 ijms-24-09182-f008:**
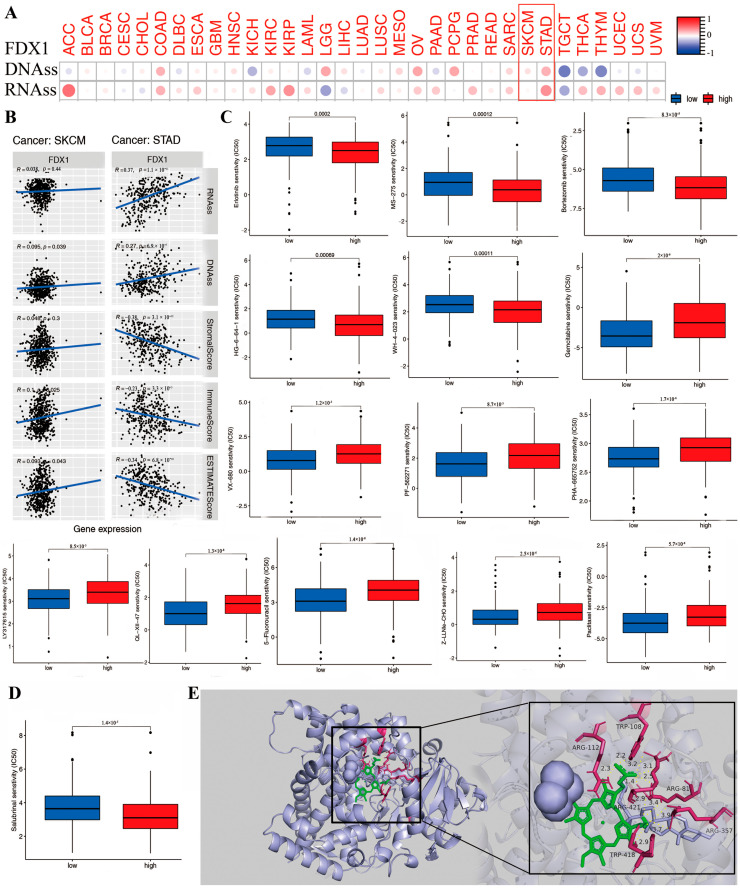
The correlation between FDX1 expression and anti-tumor drug sensitivity in different human tumors. We analyzed the association of FDX1 expression with tumor stemness of DNA/RNA (**A**,**B**). The difference of anti-tumor drug sensitivity based on IC50 value (**C**,**D**). We performed the molecular docking of FDX1 protein and salubrinal, erlotinib, gemcitabine, defactinib, WH-4-023, everolimus, chelerythrine, eleclomol, Z-LLNLE-CHO, 5-fluorouracil, and ifosfamide. These drugs had stable binding energy and hydrogen bonds between FDX1 and different anti-tumor drugs gemcitabine shown by the total view and detailed view of FDX1 (**E**). Red box are to highlight the two tumors.

## Data Availability

The authors certify that all the original data in this research could be obtained from public databases. Other data used to support the findings of this study are included within the Appendix A. All the raw data of this study are available from the first author or corresponding author upon request.

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
