# Peer review of "A Novel Oncogenic Role of FDX1 in Human Melanoma Related to PD-L1 Immune Checkpoint"

_ijms, 2023, doi:10.3390/ijms24119182_

Round 1

Reviewer 1 Report

Comment 1:  This study shows A novel oncogenic role of FDX1 by regulating Pd-L1 immune checkpoint in human melanoma. TCGA data analysis are nicely shown and described. The observations may help in further research of  PDX1 and Pd-L1 in combination with molecular targeting immunotherapy.

Comment 2: The author should provide all western blot figures with a complete protein marker ladder.

Comment 3:  Similar studies published by Chi Zhang in 2022. They also shown PDX1 in PAN cancer through TCGA analysis. Can authors explain briefly what major significant difference from Zhang article except for Pd-L1?

Comment 4: Instead of cBioPortal and GEPIA2 can the author compare TCGA data analysis for PDX1 with UALCAN cancer data analysis portal?

Minor suggestions

Comment 5:  Recheck and correct the figure number in the result section.

I cannot judge for English language but their discussion part is too lengthy

Reviewer 2 Report

1. With siRNA control for FDX1, how much is cellular normal PDL1 affected for that? Does the control affect PDL1? Figure 6 doesnt show that clearly.

2. What happens to FDX1 levels when you perform siPDL1 ? 

3. How many times the siRNA exp was repeated ? Were there 3 biological at different passages?

4. It requires more experiments to prove relation with FDX1 and PDL1. Its not convincing and clarify above mentioned points? 

5. In fig 6c, the MFI difference in not highly significant. Clarify with showing the absolute number of PDL1+ cells with FC ?

6. What is the reason for showing Fig 6D, cant understand the point. Provide a higher mag image with arrows to show whats the relevance?

7. Replot Fig 6E, text not visible.

8. In fig 5I, whats the difference between both GSE groups? Why do you see such opposite trends when predicitng survival by co-relating FDX1?

Small grammatical errors, nothing major.

Round 2

Reviewer 2 Report

In reference to the author response :

For point1 and 2 - The experiment with control siFDX1 showing baseline PDL1 expression is required. In addition control siPDL1 knockdown showing FDX expression is also required. I am not convinced there is a link for FDX1 and PDL1. Its need more validation in-vitro. If incase we go further, the title should change and reflect its only bioinformatics analysis which shows that relationship to a certain extent. 

For point 5 - The absolute number FACS difference is too less, to make conclusions with invitro experiments.

For Point 6 - Need high mag image ? cant see anything, with DAB staining ?

English language is fine.

Round 3

Reviewer 2 Report

For IHC figure, what you have provided is still not clear. Provide the following:

1)Provide a 40X image

2) IgG control staining of that same section. In about 50% of arrows I dont see any staining. Is it real staining or just artifacts.

Fine
